# UPDATE trial: investigating the effects of ultra-processed versus minimally processed diets following UK dietary guidance on health outcomes: a protocol for an 8-week community-based cross-over randomised controlled trial in people with overweight or obesity, followed by a 6-month behavioural intervention

Samuel Dicken ,[1] Janine Makaronidis,[1,2,3] Chris van Tulleken,[4] Friedrich C Jassil,[1,2,3] Kevin Hall,[5] Adrian Carl Brown [1,2,3] Claudia A.M Gandini Wheeler-Kingshott,[6,7,8] Abigail Fisher,[9] Rachel Batterham[1,2,3]

For numbered affiliations see end of article.

**Correspondence to**
Samuel Dicken;
samuel.dicken.20@ucl.ac.uk

## ABSTRACT

**Introduction** Obesity increases the risk of morbidity and mortality. A major driver has been the increased availability of ultra-processed food (UPF), now the main UK dietary energy source. The UK Eatwell Guide (EWG) provides public guidance for a healthy balanced diet but offers no UPF guidance. Whether a healthy diet can largely consist of UPFs is unclear. No study has assessed whether the health impact of adhering to dietary guidelines depends on food processing. Furthermore, our study will assess the impact of a 6-month behavioural support programme aimed at reducing UPF intake in people with overweight/obesity and high UPF intakes.

**Methods and analysis** UPDATE is a 2×2 cross-over randomised controlled trial with a 6-month behavioural intervention. Fifty-five adults aged ≥18, with overweight/obesity (≥25 to <40 kg/m$^2$), and ≥50% of habitual energy intake from UPFs will receive an 8-week UPF diet and an 8-week minimally processed food (MPF) diet delivered to their home, both following EWG recommendations, in a random order, with a 4-week washout period. All food/drink will be provided. Participants will then receive 6 months of behavioural support to reduce UPF intake. The primary outcome is the difference in weight change between UPF and MPF diets from baseline to week 8. Secondary outcomes include changes in diet, waist circumference, body composition, heart rate, blood pressure, cardiometabolic risk factors, appetite regulation, sleep quality, physical activity levels, physical function/strength, well-being and aspects of behaviour change/eating behaviour at 8 weeks between UPF/MPF diets, and at 6-month follow-up. Quantitative assessment of changes in brain MRI functional resting-state connectivity between UPF/MPF diets, and qualitative analysis of the behavioural intervention for feasibility and acceptability will be undertaken.

**Ethics and dissemination** Sheffield Research Ethics Committee approved the trial (22/YH/0281). Peer-reviewed journals, conferences, PhD thesis and lay media will report results.

**Trial registration number** NCT05627570

---

## STRENGTHS AND LIMITATIONS OF THIS STUDY

⇒ This protocol outlines the methodology for a free-living study comparing the health effects of minimally processed food and ultra-processed food (UPF) diets following EWG dietary recommendations, and the effectiveness of a 6-month behavioural support programme to reduce UPF intake.

⇒ Strengths of the randomised controlled trial include the long 8-week duration of each intervention diet, and provision of all food and drink to participants' homes to provide.

⇒ Strengths of the behavioural support programme include the use of evidence-based behaviour change techniques, and qualitative and quantitative analyses using a theoretical framework.

⇒ A subset of participants will undergo MRI brain scans.

⇒ Limitations include the exclusion of individuals with type 2 diabetes or with dietary restrictions (eg, vegan, vegetarian, halal and kosher).

## INTRODUCTION

Obesity is a global healthcare challenge. Nearly two-thirds of UK adults now live with overweight or obesity,[1] increasing the risk of life-limiting disease and premature mortality.[2] The overall annual cost of obesity in the UK is estimated at £58 billion.[3] Of which, the National Health Service (NHS) spends over £6 billion/year on obesity-related health issues, which is expected to rise to nearly £10 billion/year by 2050.[4]

A major driver of the obesity epidemic has been the shift in the food environment with increased availability of ultra-processed food (UPF).[5–7] Defined by NOVA (not an acronym), UPFs are industrial formulations with five or more ingredients, using extracts of original foods with preservatives, flavourings and colours.[8] UPFs include breakfast cereals, sweets, packaged breads and ready meals. Over 50% of UK energy intake now comes from UPFs, displacing more traditional minimally processed food (MPF).[9] In prospective cohort studies, higher UPF intakes are associated with increased risks of weight gain, overweight/obesity,[10 11] cardiometabolic disease,[12] gastro-intestinal disorders, cancer,[13] mental health problems,[14] lower physical strength[15] and all-cause mortality.[16]

The Eatwell Guide (EWG) provides the UK public with guidance for a healthy, balanced diet.[4 17 18] The EWG focuses on macronutrients and food groups, such as choosing foods lower in saturated fat, sugar and salt, and eating five daily portions of fruit and vegetables.[18] The EWG recommends 'reduced fat' or 'lower sugar' reformulated foods, based on European Food Safety Authority nutrition claim regulations. However, this promotes the intake of UPFs.[19] Given their improved nutritional quality from reduced saturated fat, added sugar and/or salt, reformulated UPFs are often marketed as 'healthy'.[20] However, the associations between high UPF intakes and adverse health outcomes appear to independent of diet quality or diet pattern.[16]

In the only randomised trial to date examining the effect of energy-matched UPF versus MPF diets, participants gained nearly 1 kg body weight on the 2-week UPF diet, but lost nearly 1 kg on the 2-week MPF diet, with over 500 kcal/day differences in energy intake.[21] The diets were also matched for presented energy density, carbohydrate, sugar, fat, sodium and fibre content. The MPF diet also led to favourable changes in appetite-regulating gut hormones compared with the UPF diet. In epidemiological substitution modelling, replacing UPFs for MPFs leads to weight loss, whereas replacing MPFs for UPFs leads to weight gain, independent of diet pattern (by adjusting for adherence to a Mediterranean diet).[22] However, to date, UK organisations such as the British Nutrition Foundation and the Scientific Advisory Committee on Nutrition do not recommend the inclusion of UPF in dietary guidelines. This is due to the largely observational body of evidence regarding the negative impact of ultraprocessing on health and lack of high-quality interventional evidence.[23 24]

**Table 1** UPDATE trial design based on PICOT

| | Adults (UCLH staff) living with overweight or obesity, and with ≥50% of habitual energy intake from UPFs |
|---|---|
| Population | |
| Intervention | A minimally processed diet complying with UK EWG guidance |
| Comparator | An ultra-processed diet complying with UK EWG guidance |
| Outcome (primary) | Mean difference in percent weight change between intervention and comparator at 8 weeks |
| Timepoints | Primary outcome assessed from baseline to 8 weeks follow-up on each diet |

EWG, Eatwell Guide; UCLH, University College London Hospitals; UPDATE, Investigating the effects of Ultra-Processed versus minimally processed Diets following UK dietAry guidance on healTh outcomEs; UPF, ultra-processed food.

Adopting a healthy diet requires a significant behaviour change and lifestyle modification which can be challenging for many individuals. The prevalence of obesity in the UK is also present among UK healthcare professionals (HCPs).[1 25] Many HCPs consume an unhealthy diet,[26] attributed to the high availability and accessibility of ready-to-eat unhealthy food and lack of healthier choices in the workplace, especially during night shifts, stress-driven eating, long working hours and staff shortages limiting the ability to take breaks.[27] It is, therefore, important to support individuals to reduce their UPF intake, and in the process, identifying barriers and facilitators to develop effective strategies for long-term behaviour change.

### Objectives and hypotheses

To date, no studies have assessed whether the health impact of following dietary guidelines is dependent on food processing, nor whether providing behavioural support to UK adults living with overweight/obesity can help to reduce their UPF intake.

Therefore, this study aims to compare the health effects between MPF and UPF diets that follow the EWG recommendations, and the feasibility and acceptability of a behavioural support programme to reduce UPF intake and be more physically active.

The primary objective is to compare weight change between MPF and UPF diets following EWG recommendations (table 1). Secondary objectives are to compare changes in cardiometabolic, behavioural, mental and hormonal outcomes between MPF and UPF diets following EWG recommendations and explore the feasibility and acceptability of the behavioural support intervention.

We hypothesise that there will be a difference in the change in weight and other health measures between the two diets; whereby consuming an ad libitum MPF diet complying with the EWG will result in weight loss and favourable changes in cardiometabolic, behavioural,

mental and hormonal outcomes, whereas consuming an ad libitum UPF diet complying with the EWG will result in no change in weight or cardiometabolic, behavioural, mental and hormonal outcomes. If the hypotheses are supported, the results would indicate important health implications of the foods that make up the majority of the energy intake of the UK adult population and the differential impact of UPF and MPF, regardless of following current NHS dietary guidance. The results have the potential to substantially impact on dietary guidelines and the dietary management of obesity worldwide, supporting the development of regulations around the marketing and labelling of UPFs, and increasing knowledge and public awareness of the health consequences of UPFs.[28] The results are directly relevant for UK public health food policy as the interventions are based on the UK EWG. Given the similarities between UK and other national dietary guidelines, there will be important implications for guidelines worldwide as well.

## METHODS AND ANALYSIS
### Trial design and study setting
UPDATE (Investigating the effects of Ultra-Processed versus minimally processed Diets following UK dietAry guidance on healTh outcomEs) is a single-site, community-based, 2×2 cross-over randomised controlled trial (RCT) followed by a 6-month behavioural intervention, conducted in the UK by the Centre for Obesity Research, Division of Medicine, University College London (UCL) and UCL Hospitals (UCLH). The protocol was designed according to Standard Protocol Items: Recommendations for Interventional Trials guidelines (https://www.spirit-statement.org).[29]

In summary, 55 adults aged 18 or older, living with overweight/obesity (between 25 kg/m$^2$ and 40 kg/m$^2$), with 50% or more of habitual energy intake from UPFs will be recruited. Participants will receive an 8-week, minimum 80% UPF diet and an 8-week, minimum 80% MPF diet, both following UK EWG macronutrient recommendations, in a random order, with a 4-week washout period. All food and drink will be provided. All participants will then receive a 6-month behavioural support programme to reduce UPF (figure 1).

The trial is funded by the National Institute for Health (NIH) and Care Research UCLH Biomedical Research Council (MRC) and Rosetrees Trust, and UCL/UCLH are the sponsors. The sponsor and funders were not involved in the design and conduct of the trial.

## METHODS: PARTICIPANTS, INTERVENTIONS AND OUTCOMES
The trial opened to recruitment on 13 January 2023, the first individual was screened on 13 March 2023, and the first randomisation on 4 April 2023. The planned end date is 31 March 2025.

### Recruitment
Potential participants will be identified through advertising at UCLH/UCL (eg, websites, Trust email, posters, internal communications) and on social media (Twitter).

Interested individuals will receive a participant information sheet (PIS) (online supplemental material 1) and be offered a phone call with the research team. Researchers will then explain the screening procedure and the aims, methods, anticipated benefits and potential hazards of the trial, and invite individuals to attend an in-person visit to undergo screening. Written informed consent will be sought a minimum of 24 hours after receiving the PIS, at the in-person visit (see online supplemental material 2 for the approved consent form). Participants can withdraw at any time, without giving a reason.

### Eligibility criteria
Written informed consent will be obtained prior to commencing screening based on the inclusion and exclusion criteria, and before any research-associated measurement is taken (table 2).

Participants will complete two non-consecutive 24-hour recalls at screening. If they meet all inclusion criteria and do not meet any exclusion criteria, they will complete a further two non-consecutive 24-hour recalls at baseline to confirm a habitual dietary intake of ≥50% UPF. The mean of all four recalls will be used to determine final habitual UPF intake. After which, they will be considered eligible.

### Interventions
Eligible participants will receive in a random order (1) an 8-week MPF diet (at least 80% MPF (NOVA group 1), <20% UPF) and (2) an 8-week UPF diet (at least 80% UPF, <20% MPF), both following EWG recommendations. During the 4-week washout between diets, participants will return to their habitual diet to minimise carryover effects.

Participants will be given all meals, snacks and drinks for each 8-week intervention to maximise adherence, ensure internal validity and minimise dropout.[30 31] Catering companies and supermarkets will deliver diets to participants' homes, two times per week. Deliveries will be scheduled according to participants' convenience.

Both diets will follow EWG recommendations, choosing foods with green or amber front of package traffic lights for total fat, saturated fat, sugar and salt over foods with red traffic lights, consuming five portions per day of fruit and vegetables, and eating a variety of foods in the right proportions.[18]

Diets will be matched for and aim to follow government recommended intakes for a 2000kcal/day diet[4 18] (table 3). Provided diets will be scaled up to 4000 kcal/day, to ensure participants are not energy restricted and energy intake is ad libitum.

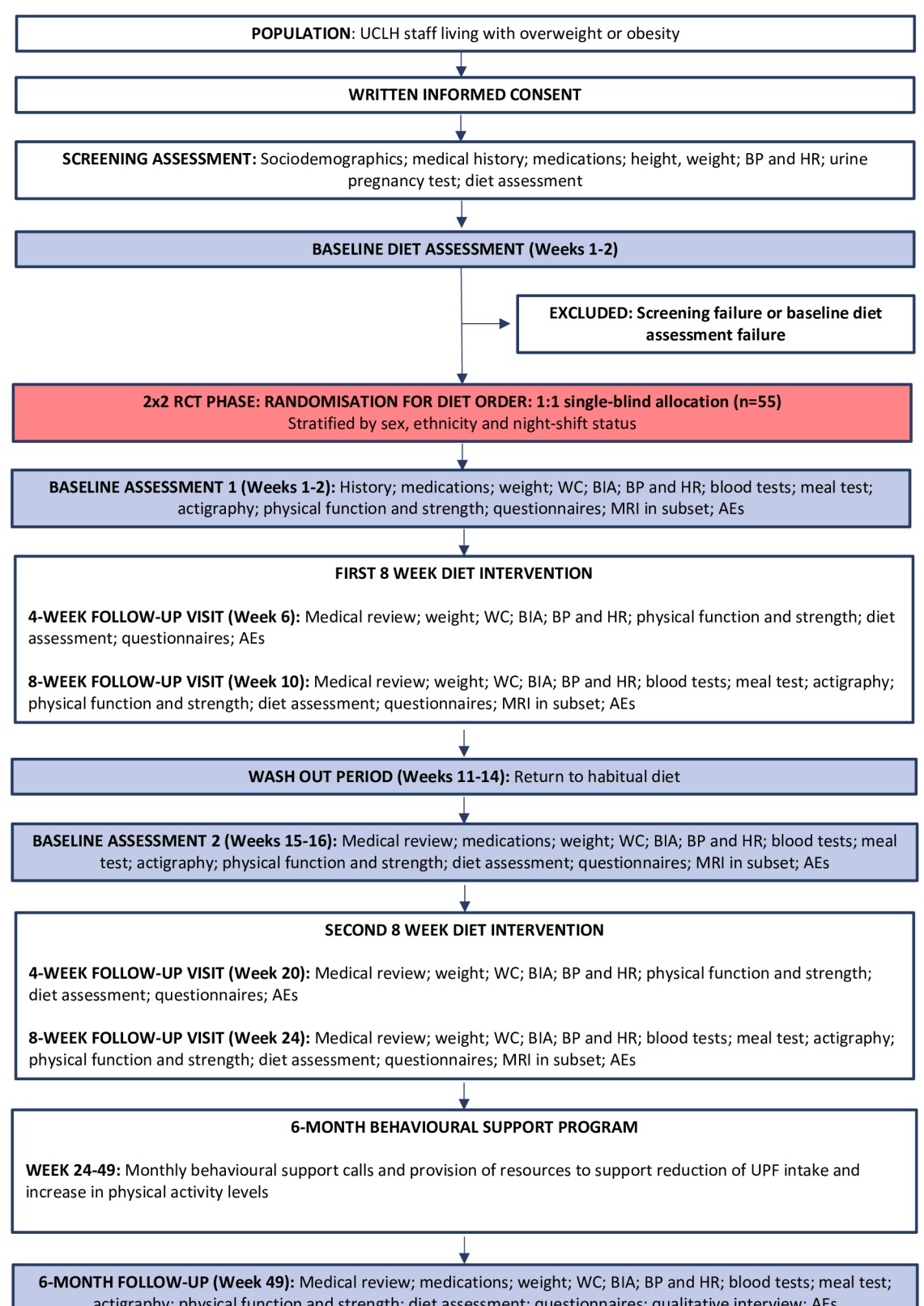

**Figure 1** Flow chart of UPDATE trial. AE, adverse events; BIA, bioimpedance analysis; BP, blood pressure; HR, heart rate; RCT, randomised controlled trial; UCLH, University College London Hospitals; UPDATE, Investigating the effects of Ultra-Processed versus minimally processed diets following UK dietary guidance on health outcomes; UPF, ultra-processed food; WC, waist circumference.

**Table 2** Eligibility criteria

| Inclusion criteria | Exclusion criteria |
|---|---|
| Staff at UCLH | Contraindication for dietary intervention |
| Adults aged ≥18 years | Participation in another clinical intervention trial |
| BMI ≥25 kg/m² (living with overweight or obesity) | Concomitant recent usage of medications that cause weight gain or weight loss |
| Weight stable (≤5% variation in body weight over preceding 3 months) | Cardiometabolic comorbidities (eg, diabetes, on insulin) |
| Have a habitual dietary intake consisting of ≥50% UPF intake as % total daily energy intake | Coeliac disease |
| Able to read and write in English | Inflammatory bowel disease |
| Medically safe to participate in a dietary intervention programme | A diagnosed eating disorder |
| Willing and able to give written informed consent | Planning a weight management programme in the next 3 months |
| Able to attend the relevant in person and remote sessions | Any diagnosed food allergy, or other allergies which limit the ability to adhere to the intervention diet |
| Able to comply with the study protocol (including dietary recommendations for each intervention and reporting adherence) | Dietary restrictions (eg, vegan, vegetarian, halal or kosher) which limit the ability to adhere to the interventions |
| Females of childbearing potential and males agree to use an effective method of contraception from the time consent is signed until the end of the intervention period and final follow-up assessment. Effective methods of contraception acceptable for UPDATE are outlined in the online supplemental material 3 | BMI >40 kg/m² or basal metabolic rate ≥2300/kcal/day (to ensure intervention diets are at least 300kcal/day greater than maintenance energy needs, based on excess energy intakes reported in Hall et al[21] |
| Females of childbearing potential must be on highly effective contraception and have a negative pregnancy test | Females who are pregnant, breast feeding or intend to become pregnant |
| | A history of drug or alcohol abuse |
| | Any other factor making the participant unsuitable in the view of investigator |

BMI, body mass index; UCLH, University College London Hospitals; UPDATE, Investigating the effects of Ultra-Processed versus minimally processed Diets following UK dietAry guidance on healTh outcomEs; UPF, ultra-processed food.

The menus will be designed to be representative of UK diets, by identifying the most commonly consumed foods in the UK National Diet and Nutrition Survey.[32] Practical and logistical aspects including price, best-before dates, storage requirements and accessibility will be factored into the design. Diets will be matched where possible, with MPF/UPF versions of the same meals (eg, Caesar salad, salmon and potatoes, porridge/breakfast cereal). UPF items are typically foods obtained from supermarkets, whereas MPFs are freshly made culinary preparations. The menu will vary across the week to prevent participant boredom and sensory-specific satiety,[33] but repeating each week to allow consistency and reduce burden. Alcohol will not be provided. Participants will be asked to keep alcohol consumption within government guidelines (≤14 units per week).[34] Participants will be educated on the EWG, but further lifestyle guidance will not be given. Menu guides will be provided with instructions and

**Table 3** Eatwell Guide dietary recommendations[4 18]

| Component | Guidance |
|---|---|
| Total fat | 35% of provided energy intake or below, 78 g or less/2000 kcal |
| Saturated fat | 10% of provided energy intake or below, 24 g or less/2000 kcal |
| Salt | Less than 6 g per day. Intake in line with current UK average intakes,[94] aiming for below 6 g per day |
| Carbohydrate | Around 50% of provided energy intake, 267 g/2000 kcal |
| Total sugars | Less than 90 g/2000 kcal, less than 18% of total energy |
| Protein | Around 15% of provided energy intake, 45 g/2000 kcal |
| Fibre | At least 18 g/2000 kcal |
| Fruit and vegetables | Five portions per day |

pictures to prepare each meal. As in previous ad libitum feeding trials investigating weight change,[35] participants will be asked to consume as much or as little of the provided diets as they desire, but they will not be asked to actively reduce their energy intake.

Participants will receive weekly calls from the research team, including dietitians, to discuss any issues with the diets and to promote adherence. Participants will have a food diary to track adherence and to record any food consumed off diet. The 4-week and 8-week diet assessments will also be used for monitoring adherence. Participants will be encouraged to report any deviations from the provided diets during weekly calls and in the food diary and to be as honest as possible, with no repercussions. Previous studies indicate high adherence when all food is provided and delivered to participant homes, where eating foods provided is more of a challenge than eating foods off the diet.[30] Minor modifications to the intervention that do not alter the overall design will be acceptable for enabling adherence. For example, adding herbs or spices (without sugar/salt) to taste or providing alternative items to facilitate food preparation if participants do not have access to a microwave (eg, swapping a microwaveable meal for a ready-to-eat meal).

### Behavioural intervention

The study is powered for the cross-over RCT, rather than for the behavioural support intervention. Therefore, in line with the MRC guidance for developing complex interventions,[36] the aim of this secondary aspect of the study is to explore the feasibility and acceptability of the behavioural support intervention to reduce UPF (primary target behaviour) and increase physical activity (PA) (secondary target). Full details of the intervention development and content will be provided in a future publication focused on the behavioural support intervention. The exit interview guide is provided in online supplemental material 4. However, in brief, the intervention was developed following the steps outlined in the Behaviour Change Wheel framework for intervention development,[37] which incorporates the 'Capability, Opportunity, Motivation' model for understanding behaviour (COM-B), augmented with Theoretical Domains Framework (TDF) for using behaviour change theory.[36 38] Qualitative studies of barriers and facilitators for dietary change in healthcare workers were extensively reviewed, with findings then mapped onto the TDF, and behaviour change techniques (BCT) consistently associated with successful diet and PA change drawn from meta-analyses of RCTs.[39–41] The intervention content combines monthly video/telephone behavioural support calls with print/online resources, based on TDF domains. Some examples of TDF domains (linked to the intervention components) are knowledge (information on what UPFs are and how to recognise them), environment (mapping of food outlets local to participant's workplace to for quickly accessible MPF options), beliefs about consequences (information about links between UPF and health), behavioural

regulation (support with goal-setting, action-planning, self-monitoring, habit) and social influence (moderated online peer support sessions). The calls will be individually tailored based on participant demographics/work patterns, responses to their baseline COM-B questionnaire, baseline levels of UPF intake and reported PA (measures of these are all described in the outcomes section) and personal barriers and motivations to change.

### Outcomes

Outcome measures will be collected at seven in-person visits. See figure 1 and online supplemental material 5 for the schedule of assessments.

The primary outcome is the mean difference in percent weight change (%WC) at 8 weeks from baseline between MPF and UPF diets. This outcome is currently being used clinically in weight management clinics and used for all NHS weight management programmes. Clinically significant weight change can occur from short-term dietary interventions and directly relates to improvements in cardiometabolic risk factors (eg, blood pressure, blood glucose, glycated haemoglobin (HbA1c), lipids),[42] physical function and quality of life.[43] Weight is an efficient and more accurate measurement to collect than other energy balance measures, such as energy intake. The hypothesised %WC in the face of the current growing obesity pandemic would have clinically relevant impacts on halting the rate of progression of obesity prevalence and mitigating the adverse impacts of adiposity-related disease, simply through shifts in the types of processed in foods being consumed.

Secondary outcomes include changes in dietary intake, waist circumference, body composition (fat mass, fat-free mass), heart rate, blood pressure, cardiometabolic risk factors (comorbidities and blood biomarkers including liver function, lipid profile, glucose, HbA1c and C reactive protein), fasted and fed metabolomics, appetite measures (fasted and fed appetite scores, circulating gut hormones and adipocytokines), sleep, sleep quality, PA levels, physical function (walking distance, leg strength, handgrip strength), mental health, quality of life, well-being and aspects of eating behaviour between MPF and UPF diets, and after the 6-month behavioural intervention compared with the first baseline assessment. In a subset of participants, changes in brain functional resting state connectivity between MPF and UPF diets will be assessed. Further behavioural intervention secondary outcomes include understanding the experiences of UPF versus MPF diets, changes in behaviour regulation and shopping expenditure from the first baseline assessment, and barriers and facilitators to reducing UPF intake.

Meal tests with blood samples for biomarkers, metabolomics, gut hormones and adipocytokines will be collected at five visits: baseline visits and 8 weeks follow-ups for the RCT and at 6 months follow-up for the behavioural intervention. A random subset of participants will undergo MRI scans at four visits: baseline visits and 8 weeks follow-ups for the RCT. The COM-B behaviour change

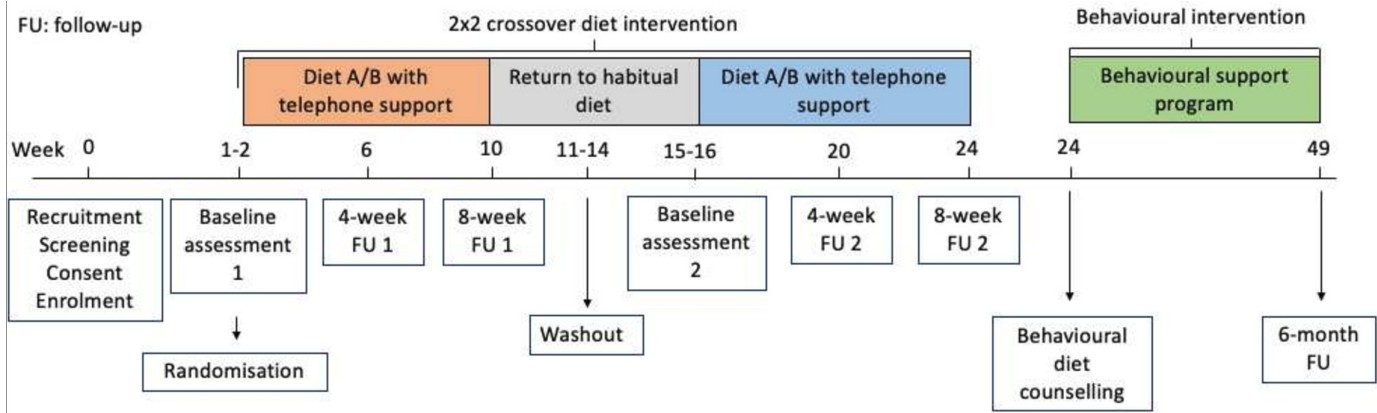

**Figure 2** Participant timeline.

questionnaire will be assessed at two visits: the first baseline visits and at 6 months follow-up.

## Participant timeline
The trial duration per participant is 49 weeks (see figure 2). The overall trial is expected to last 24 months. Recruitment commenced in March 2023 and is expected to continue until March 2024 or when 55 participants have been recruited.

## METHODS: ASSIGNMENT OF INTERVENTIONS
### Randomisation
Enrolled participants will be block randomised (stratified by night shift status, sex and ethnicity as potential treatment modifiers) to receive: (1) the MPF diet then UPF diet or (2) UPF diet then MPF diet. Randomisation will be performed using Sealed Envelope by the research team (https://www.sealedenvelope.com).

The trial statistician, but not researchers, will be blinded to the diet assignments. To prevent bias, participants are informed that the provided interventions are healthy balanced diets made with different types of food processing. All participant communications omit the terms MPF/UPF, with diets instead being referred to generally as diet 1/A or diet 2/B. Unblinding procedures are not required.

## METHODS: DATA COLLECTION, MANAGEMENT AND ANALYSIS
### Data collection methods
#### Sociodemographics
Age, gender, ethnicity, occupation, work pattern, educational level, marital status, medication intake, alcohol consumption, smoking habits, family history of obesity, cardiovascular disease and diabetes will be collected at screening.

#### Anthropometrics and body composition
Weight will be measured using an electronic scale to the nearest 0.1 kg (Tanita DC-430MAS; Tanita, Tokyo, Japan). Body composition including fat mass, fat-free mass and bone mass will be assessed using bioelectrical impedance analysis (Tanita) at each visit. Height will be determined using a stadiometer to the nearest 0.5 cm and body mass index derived from weight and height (in kg/m$^2$). Waist circumference will be measured using a non-stretch tape measure at the iliac crest.[44]

#### Vital signs
Blood pressure will be recorded in triplicate, seated, alongside heart rate with an automated sphygmomanometer and oximeter.

#### Mixed meal test and blood samples
Mixed meal tests will be used to assess circulating gut hormones, adipocytokines and metabolomics, given the significant differences in metabolites and appetite hormones between UPF and MPF diets in.[21 21] Participants will consume 187.5 mL Ensure Compact 2.4 kcal (https://nutrition.abbott/uk/product/ensure-compact) after an overnight fast. Fasted blood samples taken before the test include venous glucose, HbA1c, liver function tests, lipid profile and C reactive protein, relevant for cardiometabolic health. Blood samples and subjective appetite assessments using a Visual Analogue Score will be collected immediately before, and 15 and 30 min after starting the liquid meal.[45] Fasted and postprandial circulating appetite hormones (ghrelin, peptide YY and glucagon-like-peptide-1), adipocytokines as markers of adipose tissue inflammation (leptin, resistin, adiponectin, interleukin-6 and tumour necrosis factor alpha) and metabolomics analysis of 250 metabolites using the Nightingale platform (https://nightingalehealth.com) will be collected and stored at −80°C.

#### PA levels and sleep quality
PA levels will be measured objectively using accelerometry (ActiGraph wGT3X-BT). The device has been previously used in clinical research given its practicality, non-invasiveness, and reliability and accuracy in measuring PA in free-living adults.[46] Data recorded includes energy expenditure, metabolic equivalents and sleep activity. The short-form International Physical Activity Questionnaire (IPAQ-SF) will be used to

subjectively assess light, moderate and vigorous PA, and time spent sitting.[47–49] The IPAQ-SF contains seven questions about PA in the past 7 days.[50] ActiGraph wGT3X-BT will also objectively measure sleep quality and quantity. The Pittsburgh Sleep Quality Index (PSQI) is a validated 19-item measure for determining sleep quality in the past month, to distinguish between good and poor sleepers.[51]

### Physical function and strength

Static muscle strength of the upper extremities will be assessed using a handgrip dynamometer (Jamar Hydraulic Hand Dynamometer, Patterson Medical). Three measures will be recorded (in kg) with each hand, alternately, while seated. Functional capacity will be assessed using the 6 min walk test, a self-paced submaximal assessment, validated in people living with obesity.[52] The test will be performed according to the '*American Thoracic Society Statement: Guidelines for the Six-minute Walk Test*' protocol.[53] The pretest and post-test heart rate, total distance covered, perceived physical exertion and any physical problems will be recorded. Lower body functional capacity will be assessed using the sit-to-stand test.[54] Participants will be requested to perform five sit-to-stand repetitions as fast as possible with arms crossed over the chest. The number of repetitions and time to completion will be recorded.

### Quality of life, mental health and well-being questionnaires

EuroQol 5-Dimensions 3-Levels (EQ-5D-3L) is a 6-item questionnaire containing a descriptive system assessing five domains: mobility, self-care, usual activities, pain/discomfort and anxiety/depression using three levels (eg, mobility: I have no problems walking about, I have some problems walking about or I am confined to bed) and a visual analogue scale reporting self-rated health from 0 to 100.[55] Impact of Weight on Quality of Life-Lite (IWQOL-Lite) is a 31-item, obesity/overweight-specific measure of health-related quality of life relating to five domains: physical function, self-esteem, sexual life, public distress and work. Higher scores indicate a better quality of life.[56] The Warwick Edinburgh Mental Well-being Scale (WEMWBS) is a 14-item, positively worded, validated measure of mental well-being. Five responses are summed for a total score between 14 and 70, with higher scores indicating greater positive mental well-being.[57] The Patient Health Questionnaire (PHQ-9) is a nine-item validated measure to assess the severity of depression.[58] Each item is rated from 0 to 3, for a total score out of 27. A score of 1–4 indicates minimal depression, 5–9 indicates mild, 10–14 indicates moderate, 15–19 indicates moderately severe, with 20–27 indicating severe depression. The Generalised Anxiety Disorder assessment (GAD-7) is a seven-item anxiety scale to measure the severity of GAD.[59] Each item is rated from 0 to 3, for a total score out of 21. A score of 11–15 indicates moderately severe anxiety, with 16–21 indicating severe anxiety.

### Eating Behaviour Questionnaires

The Power of Food (PoF) scale is a 15-item validated measure to assess the psychological impact of living in food-abundant environments.[60] The Control of Eating Questionnaire (CoEQ) is a 21-item validated measure of the severity and type of food cravings.[61]

### Behaviour Change Questionnaire

Participants will complete a questionnaire investigating the barriers and facilitators to PA and healthy eating based on COM-B.[62] The PA component contains 137 items, and the eating component contains 120 items.

### MRI

Several aspects of UPFs may influence eating behaviour, including changes to the food composition, food matrix degradation and behavioural aspects, including large portion presentation, cost, availability, shelf life, heavy marketing and attractive packaging.[16] UPFs have been suggested to have addictive-like properties,[63] evoking strong emotional reactivity from visual cues.[64] Functional brain MRI has provided valuable insights into appetite regulation,[65] and how obesity impacts the neurobiology of weight regulation. However, the neurobiological impact of high-UPF diets is largely unknown. A subset of participants (n=24) will undergo an advanced MRI brain protocol at the baseline and 8-week visits of the RCT (four visits), to assess a number of changes in brain properties: (1) functional resting-state connectivity between regions implicated in eating behaviour, metabolism and swelling linked to diffuse inflammation, and their reversibility[66]; (2) microstructure brain properties extracted from diffusion weighted imaging data[67]; (3) changes in brain dynamics, based on each subject's functional and structural connectivity[68]; (4) metabolite changes, such as glutamate, N-acetyl aspartate and Inositol, through brainstem MR spectroscopy[69]; (5) regional volume changes through high-resolution structural scans[66] and (6) relaxometry properties alterations affected by inflammation.[70]

### Assessment of dietary adherence and intake

In line with previous trials,[71] multiple approaches will be used to assess dietary intake and adherence, including 24-hour recalls, food frequency questionnaires (FFQ) and image-based dietary assessment (IBDA).

Intake24[72] is a validated, online, self-reported 24-hour recall system, based on a multiple-pass recall suitable for the general population[73 74] (https://intake24.co.uk). The web-based recall method is convenient, efficient and ensures coding consistency. Participants enter all food and drink consumed in the previous 24 hours (from waking up to going to sleep) into Intake24 on two non-consecutive days per visit period. Each recall takes roughly 12 min to complete.[75] The first recall at screening will be conducted with the research team to ensure adequate training. Links will then be sent to participants to complete further recalls remotely. Intake24 is connected to the National Diet and Nutrition Survey

**Table 4** Time points of questionnaires in the UPDATE trial

| Visit | No of questionnaires | Time to complete |
|---|---|---|
| Screening | Intake 24 only | 12 min |
| RCT:<br>► First baseline visit<br>Behavioural intervention:<br>► 6 months follow-up<br>(visits 2 and 9) | All 12 questionnaires | 90 min |
| RCT:<br>► Week 4<br>► Week 8<br>► Second baseline visit<br>(visits 3, 4, 5, 6 and 7) | 11 questionnaires (all except the COM-B healthy eating and PA questionnaire) | 60 min |

COM-B, Capability, Opportunity, Motivation-Behaviour; PA, physical activity; RCT, randomised controlled trial; UPDATE, Investigating the effects of Ultra-Processed versus minimally processed Diets following UK dietAry guidance on healTh outcomEs.

Nutrient Databank to provide nutrient outputs and has been coded into NOVA by the research team for calculating UPF/MPF intake.[76] Intake24 includes details on the brand names of products, facilitating assignment of NOVA groups.

The European Prospective Investigation into Cancer and Nutrition (EPIC)-Norfolk FFQ is a validated, semiquantitative measure of average dietary intake over the past year.[77–79] The FFQ contains a 130-item food list followed by detailed questions regarding the food list. For each food item, participants tick the most appropriate frequency of consumption of that item from nine options (from never or less than once per month, to 6+ per day).

Participants will complete an FFQ and two nonconsecutive day 24-hour dietary recalls at baseline, and 4 weeks and 8 weeks for both diets and at the 6-month behavioural intervention follow-up (figure 1).[80] IBDA will also be completed on two non-consecutive days during weeks 4 and 8 of the RCT, whereby participants will take photos before and after their ad libitum meals/snacks.

### Questionnaires summary
There are 12 questionnaires/surveys: PSQI, IPAQ-SF, EQ-5D-3L, IWQOL-Lite, WEMWBS, PHQ9, GAD7, PoF, CoEQ, COM-B healthy eating and PA questionnaire, Intake24 and EPIC-Norfolk FFQ. Time points for completion are detailed in table 4 and in online supplemental material 5.

### Feasibility/acceptability and process evaluation of the behavioural intervention
Feasibility/acceptability and process evaluation will be described in full in the subsequent paper focused on the behavioural intervention. However, the main feasibility/acceptability outcome will be the percentage of intervention calls successfully delivered, as well as retention to the study. For quality control intervention calls will be recorded and coded against a checklist to ensure that target BCTs are being delivered as planned. Acceptability of the intervention content and process will be gathered in qualitative interviews. Changes in other outcomes pre

and post the behavioural intervention will be explored as secondary.

### Qualitative interviews
In the first behavioural intervention call, before being told that the intervention(s) focus on processing, participants will be asked to describe their experiences of each of the trial diets 1A/2B (positives and negatives, how they felt physically/mentally, barriers they faced to adherence) then probed on which diet they thought was UPF versus MPF and why. In addition, at the end of the behavioural intervention, participants will be invited to take part in a one-to-one semistructured telephone interview about their experiences of the trial overall, motivations for participation, barriers to reducing UPF and increasing MPF, and experience of the behavioural intervention specifically (whether they found materials and process acceptable/useful).

### Data methods: monitoring
#### Data monitoring
Data generated from this trial will be handled (including collection, storage, processing and disclosure) in accordance with all applicable legal and regulatory requirements, including the UK Data Protection Act (DPA) 2018 and European Union General Data Protection Regulation (EU GDPR 2016/679). All members of the research team are trained in information governance and research integrity.

All data storage mediums will comply with the NHS Information Governance Toolkit. All physical data will be stored in a secure room, with limited access only to members of the research team. All computers storing electronic data will be encrypted and password protected. Data will be first stored on paper case report forms (CRF), securely kept in a locked cabinet in a locked office and then recorded electronically on a purpose-built trial database (eCRF) using the REDCap system (https://www.project-redcap.org). REDCap is hosted within the UCL Data Safe Haven, a ISO27001 certified secure environment protected by dedicated firewalls, accessed by

role-based user accounts with multifactor authentication. Each participant has their own eCRF, listed under their participant identification number (PIN). Questionnaires will be provided electronically and stored directly in REDCap. The REDCap database contains range validity checks with warnings for erroneous or missing values, and initial data entry from the paper CRF will be verified by a second team member for quality assurance. The REDCap application also provides a comprehensive audit trail showing all changes to the data.

Data confidentiality is outlined in the PIS (online supplemental material 1). Participant initials and their PIN will be used on records. The CRF will not bear the participant's name or other personal identifiable data. Identifiable information will be stored in a separate electronic database to the database containing trial data. Participants' personal identity and data collected in the study cannot be connected by anyone outside the study team. The data generated from this trial will not be transferred to any party not identified in the protocol and will not be processed and/or transferred other than in accordance with the participants' consent. Confidentiality will be maintained by user agreements prior to access, ensuring all researchers uphold the principles of Good Clinical Practice (GCP), EU GDPR 679/2016 and DPA 2018.

### Retention

Loss to follow-up will be minimised through weekly telephone calls during the RCT and monthly calls during the 6-month behavioural intervention to discuss any issues and adherence, encouraging participants to discuss any difficulties with researchers. Other procedures will be adopted for maintaining participation in the study (eg, reminders before scheduled trial visits, sending greetings messages, personalising letters and keeping measures short in terms of completion time). No data will be collected after withdrawal.

### Sample size

The sample size of 55 is based on[21] showing 0.9 kg weight loss following a 2-week MPF diet, with an SD of the mean difference in weight change of 1.98 kg between MPF and UPF diets (mean: 1.85 kg), the predicted weight loss if the MPF diet were continued for 8 weeks (using the NIH bodyweight planner (https://www.niddk.nih.gov/bwp)).

No interventional studies to date have compared diets differing in the nature, extent and purpose of food processing over this duration. To best estimate the expected difference in %WC over this time frame, the results from the only controlled feeding trial comparing UPF and MPF diets matched for diet quality on weight change were used,[21] which lasted for 2 weeks. Inclusion criteria for UPDATE require participants to have a habitual diet high in UPF, thus the UPF diet is hypothesised at best to lead to no change in weight. Whereas, the MPF diet will result in weight loss, with a similar initial trajectory to that in Hall *et al.*[21] The NIH bodyweight planner takes into account that weight change is not linear based on the initial calorie deficit and tends to plateau and considers key factors that influence metabolic rate. The mean age, height, weight and PA levels of the male and female participants from Hall *et al* were entered into the bodyweight planner to produce a quantitative estimate of weight change on the MPF diet over 8 weeks, which was converted percentage weight change. Male participants in Hall *et al*[21] would achieve 2.8% (2.2 kg) weight loss after 8 weeks on the MPF diet, and female participants would achieve 2.7% (2.1 kg) weight loss. There is no clear value from the literature as to what the SD should be for the mean difference in weight change between 8 week MPF and UPF diets. The expected SD must, therefore, be estimated. The 1.98 kg value for the SD of the mean difference in weight change from Hall *et al* was used as the starting point for the expected SD after 8 weeks. This was around 1.1× the mean difference in weight change between MPF and UPF diet interventions. By assuming that the SD would increase over 8 weeks compared with 2 weeks, the SD was assumed to increase to 2× the mean difference in weight change between MPF and UPF diets. Other trials longer in duration than Hall *et al* were considered, but other trials have a number of fundamental differences to the planned trial, which limits the ability to extrapolate their findings (eg, not necessarily ad libitum or cross-over trials, testing diets that are unrelated to the concepts of MPF or UPF, or comparing an intervention to a control diet, (whereas UPDATE involves two interventions)). These generally reported SDs for the mean weight change of each group, rather than of the mean difference between groups. For each group over several weeks (eg, 12 weeks), the SDs tended to be around 0.1–0.9× the mean weight change. Therefore, the Hall *et al*'s estimate was considered most applicable, relevant and conservative.

Assuming weight loss on the MPF diet and no weight change on the UPF diet, 44 participants are required to detect a mean difference of 2.7% weight change between diets (SD of 5.4% (2× the mean difference), power=0.9, alpha=0.05, with a two-sided paired t-test, using SPSS V.27.0). With a 20% drop-out rate based on previous controlled feeding trials,[81 82] 55 participants will be recruited. This is comparable to attrition rates in previous multiarm, community-based cross-over trials lasting several months, where participants are provided with all meals.[83–86]

The target sample size for the qualitative interviews is 20–30. Data saturation is controversial, but researchers will aim for meaningful saturation, whereby further interviews produce minimal, or no changes to the coding framework and allow complete understanding of thematic codes.[87] Where possible, participants will be selected to ensure representation from across ethnicities, night shift patterns, genders, treatment allocation arms. All participants will be offered the behavioural intervention calls, so will be asked about experiences of the provided trial diets.

## Statistical methods

A Consolidated Standards of Reporting Trials (CONSORT) diagram and descriptive statistics will be used to outline the trial sample.[88] The primary outcome will be analysed using mixed-effects models, with a random effect for participants, adjusting for stratification variables and any baseline variables not balanced between arms. Age will also be included as an adjustment covariate. An intention-to-treat analysis will be conducted,[89] with values presented by randomisation group and all available data analysed as randomised. Additional models will include a per-protocol analysis, and repeated-measures analysis additionally using data from 4 week follow-ups. Bias from missing data will be dealt with using multiple imputation.[90] The 4-week washout period between diets will minimise any carryover effects,[88] with any residual effect assessed between arms. To account for non-adherence (consuming more than one meal per week off the intervention diet), inverse probability weighting will be used to reweight the remaining sample.[91] There is no planned interim analysis.

Secondary outcome variables for the RCT and behavioural intervention will be analysed using mixed-effects or linear, binary or ordinal regression models, where appropriate.

Qualitative interviews will be analysed using framework analysis, following the stages outlined by Gale *et al*.[92] Transcripts will be deductively coded, broadly mapping barriers and facilitators to the COM-B model[36 37] and TDF[38]—an approach previously used to understand the experience of novel diets,[93] as well as inductively coding aspects that may not naturally fit into these groupings. Recordings will be converted into text using transcription software. Two researchers will initially independently analyse three interviews, before meeting to discuss and develop an initial framework. Additional transcripts will then be coded based on this framework, with additional factors added as they arise. Two researchers will interpret data and write the report in accordance with the consolidated criteria for reporting qualitative research checklist (www.equator-network.org/reporting-guidelines/coreq).

## METHODS: MONITORING

A data and safety monitoring plan (DSMP) plan is in place for quality control to ensure adherence to the approved protocol (V.1.3, 26 July 2023). Charters outline the roles and responsibilities of the trial management group (TMG) and trial steering committee (TSC). The TMG includes the chief investigator (CI) (RB), trial coordinator and manager (SD) and coinvestigators/researchers (ACB, CAGW-K, AF, FCJ and JM). The TMG is responsible for maintaining and overseeing the trial, meeting before recruitment and every 3 months throughout the trial. The TMG

approved the final trial protocol and eCRF, as well as any subsequent amendments. The TSC will provide supervision and act on behalf of the funders and sponsor, recommending any appropriate amendment/actions as necessary. The TSC includes a lay member and will be chaired by an external academic member with experience leading clinical trials. The trial may be stopped before completion on recommendation from the TSC, or sponsor and CI.

A data monitoring committee will not be set up as there are no intervention-related adverse events (AEs), serious AEs (SAEs) or major risks associated with participation. Risks are considered minimal and are outlined in the PIS and CF (online supplemental materials 1 and 2). Recording/reporting AEs will commence from written informed consent until the 6-month follow-up (week 49). All SAEs will be reported to the Sponsor within 24 hours of becoming aware. Incidental findings will be reported to participants and their GP.

## Auditing

Monitoring and auditing (eg, adherence, deviations, withdrawals, AEs/SAEs) is outlined in the DSMP and will be conducted by the CI, TMG and TSC in accordance with the UK Policy Framework for Health and Social Care Research, and in accordance with the sponsor's monitoring and audit policies and procedures.

## ETHICS AND DISSEMINATION
### Research ethics approval

The Yorkshire & The Humber—Sheffield Research Ethics Committee approved the trial on 22 December 2022 (22/YH/0281), and the study was prospectively registered on ClinicalTrials.gov (NCT05627570). The trial is currently recruiting and ongoing. The trial will be conducted in compliance with the principles of the Declaration of Helsinki 1996 and the principles of the International Conference on Harmonisation GCP. Any amendments will be recorded in academic publications and will be submitted for approval to the Sponsor and Research Ethics Committee prior to implementation. Protocol V.1.3, 26 July 2023. Sponsor contact: (UCLH/UCL) Joint Research Office (uclh.randd@nhs.net). On 16 June 2023, the UPF intake inclusion criterion was amended from ≥60% to ≥50% to better reflect the average intake of participants in London, compared with intakes across the UK.[9]

### Ancillary and post-trial care

There will be no diet provisions or behavioural support given at the end of the trial. Consent will be sought for future research and the storage and future use of participant samples in the Obesity Research Biobank Syndicate.

### Dissemination

Results will be disseminated through publications, conferences and social and lay media. Progress and

analyses may be presented to the sponsor and funding bodies. Authorship will follow the International Committee of Medical Journal Editors guidelines. Participant data will not be made publicly accessible.

**Author affiliations**
[1]Centre for Obesity Research, Department of Medicine, University College London, London, UK
[2]Bariatric Centre for Weight Management and Metabolic Surgery, University College London Hospital (UCLH), London, UK
[3]National Institute for Health Research, Biomedical Research Centre, University College London Hospital (UCLH), London, UK
[4]Division of Infection, University College London, London, UK
[5]National Institutes of Health, National Institute of Diabetes and Digestive and Kidney Diseases, Bethesda, Maryland, USA
[6]NMR Research Unit, Department of Neuroinflammation, UCL Institute of Neurology, University College London, London, UK
[7]Department of Brain and Behavioural Sciences, University of Pavia, Pavia, Italy
[8]Digital Neuroscience Center, IRCCS Mondino Foundation, Pavia, Italy
[9]Department of Behavioural Science and Health, University College London, London, UK

**Acknowledgements** The authors would like to thank: UPDATE participants and collaborators, the Intake24 team (https://intake24.co.uk/), the patient and public involvement focus group, EPIC-Norfolk for use of their FFQ (https://www.epic-norfolk.org.uk/for-researchers/ffq/), REDCap (https://www.project-redcap.org/) and Out Of The Box Food. Acknowledgements also go to Caroline Buck, Gabriella Heuchan and Rana Conway, for their substantial input into the behavioural intervention development.

**Contributors** RB conceptualised the trial; SD wrote the initial manuscript and protocol draft; RB, SD, ACB, CAGW-K, KH, AF, JM, FCJ and CvT contributed to the intervention design, methodology and assessments, RB, SD, ACB, AF, JM and FCJ contributed to the study protocol.

**Funding** This work was supported by National Institute for Health and Care Research Biomedical Research Centre (NIHR BRC) grant BRC530a and Rosetrees Trust grant PGL22/100041.

**Competing interests** SD is funded by the Medical Research Council (MR/N013867/1) and receives royalties from Amazon for a self-published book that mentions ultraprocessed food, and payments from Red Pen Reviews as a contributor. RB reports honoraria from Novo Nordisk, Eli Lilly, Medscape, ViiV Healthcare and International Medical P and advisory board and consultancy work for Novo Nordisk, Eli Lilly, Pfizer, Gila Therapeutics, Epitomee Medical and ViiV Healthcare and from May 2023 is an employee and share holder of Eli Lilly and Company. ACB reports honoraria from Novo Nordisk, Office of Health Improvement and Disparity, Johnson and Johnson and Obesity UK outside the submitted work and is on the Medical Advisory Board and shareholder of Reset Health Clinics. CAGW-K receives funding from Horizon2020 (Research and Innovation Action Grants Human Brain Project 945539 (SGA3)), BRC (#BRC704/CAP/CGW), MRC (#MR/S026088/1), Ataxia UK, Rosetrees Trust (#PGL22/100041 and #PGL21/10079) and is a shareholder in Queen Square Analytics. JM is funded by the NIHR and reports funding from the NIHR BRC and the Society for Endocrinology. JM reports institutional funding from Novo Nordisk, Rhythm Pharmaceuticals and Innovate UK outside the submitted work. KH is supported by the Intramural Research Program of the National Institutes of Health, National Institute of Diabetes & Digestive & Kidney Diseases. CvT receives royalties for a book on ultraprocessed food.

**Patient and public involvement** Patients and/or the public were involved in the design, or conduct, or reporting, or dissemination plans of this research. Refer to the Methods section for further details. The trial, including the menu, assessments and participant burden have been reviewed and designed with input from NHS UCLH staff at following a focus group session before ethics submission. In addition, Obesity Empowerment Network UK members, with lived experience of obesity, have also contributed to the design.

**Patient consent for publication** Not applicable.

**Provenance and peer review** Not commissioned; externally peer reviewed. The study has been peer reviewed in accordance with the requirements outlined by UCL. The study was internally peer-reviewed by coinvestigators of the research team, and externally peer-reviewed by two peer reviewers at UCL. The protocol has been updated with additional detail following the comments provided by the peer reviewers. Furthermore, the trial was peer-reviewed as part of the funding grant application by Rosetrees Trust by external peer reviewers.

**ORCID iDs**
Samuel Dicken http://orcid.org/0000-0001-5663-1715
Adrian Carl Brown http://orcid.org/0000-0003-1818-6192

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
