## [Reviewer comments · BMJ Open]

ARTICLE DETAILS

TITLE (PROVISIONAL)	UPDATE trial: Investigating the effects of Ultra-Processed versus minimally processed Diets following UK dietary guidance on health outcomes: a protocol for an 8-week community-based crossover randomised controlled trial in people with overweight or obesity, followed by a 6-month behavioural intervention.
AUTHORS	Dicken, Samuel; Makaronidis, Janine; van Tulleken, Chris; Jassil, Friedrich; Hall, Kevin; Brown, Adrian; Gandini Wheeler-kingshott, Claudia; Fisher, Abigail; Batterham, Rachel

VERSION 1 – REVIEW

REVIEWER	Euridice Martinez Steele University of Sao Paulo, Department of Nutrition, School of Public Health
REVIEW RETURNED	19-Oct-2023

GENERAL COMMENTS	UPDATE trial: Investigating the effects of Ultra-Processed versus minimally processed Diets following UK dietary guidance on health outcomes: a protocol for an 8-week community based crossover randomised controlled trial in people with overweight or obesity, followed by a 6-month behavioural intervention- REVIEW Abstract p. 3 Line 11. 'No studies have assessed whether the health impact of dietary guidelines depends on the extent and purpose of processing, nor provided behavioural support to UK adults with overweight/obesity to reduce their UPF intake.' This sentence seems incomplete and unclear. p. 3 Line 17. 'Fifty-five adults aged ≥ 18, living with overweight/obesity (≥ 25 to < 40 kg/m²), with $\geq 50\%$ of habitual energy intake from UPFs will receive an 8-week UPF diet and an 8-week minimally processed food (MPF) diet, both following EWG recommendations, in a random order, with a 4-week washout period.' I suggest mentioning that diets will be delivered to participants' homes. p. 3 Line 23. 'The primary outcome is the difference in weight change between UPF and MPF diets.' Is the primary outcome the 'difference in weight change between UPF and MPF diets' or 'the weight change between UPF and MPF diets'? Is this at 8 weeks or at both 8 weeks and at 6-months follow-up? If at 6-months follow-up, what (measured when?) would be the initial weight used as a comparator for the 6-months weight? How and when would UPF and MPF diets be characterized?
---

p. 3 Line 24. 'Secondary outcomes include changes in diet, waist circumference, body composition, heart rate, blood pressure, cardiometabolic risk factors, appetite regulation, sleep quality, physical activity levels, physical function and strength, wellbeing and aspects of behaviour change and eating behaviour at 8 weeks between UPF and MPF diets, and at 6-months follow-up.' For 6-months follow-up assessments, what (measured when?) initial outcome assessments would be used as a comparator and how (and when?) would UPF and MPF diets be characterized?

Introduction.

p. 4 Line 37. 'In the only randomised trial to date examining the effect of energy-matched UPF versus MPF diets participants gained nearly 1kg body weight on the 2-week UPF diet, but lost nearly 1kg on the 2-week MPF diet, with over 500kcal/day differences in energy intake²¹. The diets were also matched for presented energy density, carbohydrate, sugar, fat, sodium and fibre content. The MPF diet also led to favourable changes in appetite-regulating gut hormones compared with the UPF diet. This further suggests that the adverse effects of high UPF intakes are independent of individual nutrients such as fat, salt and sugar.' Because it was the offered/presented diets that were matched for nutrients rather than the actual consumed diets, I do not think the matching can disprove that nutrients such as fat, salt and sugar played a role in the weight gain observed during the UPF diet. Did participants consume the same amount of fat, salt and sugar during the UPF and MPF diets? Only if so, might we conclude that nutrient consumption was not a driver of weight gain.

p. 4 Line 56. 'Adopting a healthy diet still presents a significant behaviour change and lifestyle modification which can be challenging for many individuals.' Should 'presents' be replaced by 'requires'?

Objectives and hypotheses

p. 5 Line 29 'We hypothesise that there will be a difference in the change in weight and other health measures between the two diets; whereby consuming an ad libitum MPF diet complying with the EWG will result in weight loss and favourable changes in cardiometabolic, behavioural, mental and hormonal outcomes, whereas consuming an ad libitum UPF diet complying with the EWG will result in no change in weight or cardiometabolic, behavioural, mental and hormonal outcomes.' Do you hypothesise that 'there will be a difference in the change in weight and other health measures between the two diets' or that 'there will be a change in weight and other health measures between the two diets'?

Why do you hypothesise that consuming an ad libitum UPF diet complying with the EWG will result in no change in weight or cardiometabolic, behavioural, mental and hormonal outcomes? Why don't you expect lower weight loss and favourable changes in cardiometabolic, behavioural, mental and hormonal outcomes instead of null effects (if ad libitum UPF diet complying with the EWG has better nutritional qualities than the UPF diet likely less compliant with the EWG before entering the trial)?

Methods and analysis

p. 9 Line 11. 'Participants will complete two non-consecutive 24-hour recalls at screening. If they meet all inclusion criteria and do not meet any exclusion criteria, they will complete a further two nonconsecutive 24-hour recalls at baseline to confirm a habitual

dietary intake of $\geq 50\%$ UPF. After which, they will be considered eligible.' How exactly will you handle the four 24-hour recalls for confirming a habitual dietary intake of $\geq 50\%$ UPF? Will means across measures be calculated?

p. 10 Line 13. 'Alcohol will not be provided. Participants will be asked to keep alcohol consumption within government guidelines (14 units per week)³⁴.' Will participants be instructed to stick their diets to the delivered foods? If so, wouldn't allowing for alcohol consumption escape from this instruction?

p. 10 Line 16. 'Menu guides will be provided with instructions and pictures to prepare each meal.' Doesn't this exclusively apply to MPF diets? If so, shouldn't this be mentioned?

p. 12 Line 32. 'Enrolled participants will be block randomised (stratified by night-shift status, sex and ethnicity as potential treatment modifiers) to receive: (1) the MPF diet then UPF diet, or (2) UPF diet then MPF diet.' Wouldn't stratification by age also make sense?

p. 15 Line 9. 'Intake24 is a validated, online, self-reported 24-hour recall system, based on a multiple-pass recall suitable for the general population^{73,74} (<https://intake24.co.uk>). The web-based recall method is convenient, efficient and ensures coding consistency. Participants enter all food and drink consumed in the previous 24 hours (from waking up to going to sleep) into Intake24 on two non-consecutive days per visit period. Each recall takes roughly 12 minutes to complete⁷⁵. The first recall at screening will be conducted with the research team to ensure adequate training. Links will then be sent to participants to complete further recalls remotely. Intake24 is connected to the National Diet and Nutrition Survey Nutrient Databank to provide nutrient outputs, and has been coded into NOVA by the research team for calculating UPF/MPF intake.⁷⁶' Does the Intake24 collect brand name information (important to determine Nova classification) and will brand name information be taken into account when assigning Nova groups?

p. 15 Line 23. 'The European Prospective Investigation into Cancer and Nutrition (EPIC)-Norfolk FFQ is a validated, semi-quantitative measure of average dietary intake over the past year⁷⁷⁻⁷⁹. The FFQ contains a 130-item food list followed by detailed questions regarding the food list. For each food item, participants tick the most appropriate frequency of consumption of that item from nine options (from never or less than once per month, to 6+ per day). Participants will complete an FFQ and two non-consecutive day 24-hour dietary recalls at baseline, and 4- and 8-weeks for both diets, and at the 6-month behavioural intervention follow-up (Figure 1)⁸⁰.' Will the same Nova coding used in previously published EPIC studies be applied to FFQ food items? If FFQ relates to dietary intake over the past year, will it be applied several times to handle potential inconsistencies?

p. 15 Line 23. 'IBDA will also be completed on two non-consecutive days during weeks 4 and 8 of the RCT, whereby participants will take photos before and after their ad libitum meals/snacks.' Why isn't IBDA also applied at baseline and at 6-month follow-up?

p. 15 Line 23. 'Questionnaires summary. There are twelve questionnaires/surveys: PSQI, IPAQ-SF, EQ-5D-3L, IWQOL-Lite,

	WEMWBS, PHQ9, GAD7, PoF, CoEQ, COM-B healthy eating and PA questionnaire, Intake24 and EPICNorfolk FFQ. Time points for completion are detailed in Table 3 and in Supplementary Material 4.' Will means across measures be calculated to obtain summary measures? p. 18 Line 35. 'Where possible, participants will be selected to ensure representation from across ethnicities, night-shift patterns genders, treatment allocation arms.' Why not age, also?
--	--

REVIEWER	Andaleeb Rahman Cornell University
REVIEW RETURNED	22-Nov-2023

GENERAL COMMENTS	Ultra-processed food are increasingly believed to be the leading cause of obesity and associated non-communicable diseases in the world. In UK, Eatwell Guide (EWG) has been prepared by the government to promote a health diet which is low in sugar and contains reduced amount of fat. Yet, the EWG guidelines do not discourage the consumption of UPFs as some of them--with better macronutrients and reduced salt and fat content--are marketed as 'healthy'. This proposed research claims to be the first randomized trial (RCT) study--the gold standard in causal methods--which compares the health effect of processed food. It aims to compare the health outcomes (weight being the primary outcome along with cardiometabolic, behavioral, mental and hormonal difference) of individuals who consumer the similar food in accordance with the EWG recommendations but the diets differ in terms of their processing levels -- Ultra-processed food (UPFs) versus minimally processed food (MPFs). I feel that the authors have an impeccable understanding of the clinical research methods proposed to be employed here. I am extremely satisfied with the protocol presented here which covers all the bases. The sampling strategy and research design seems solid and I don't think that the small sample size would be stumbling block for statistical inference here. I believe that this is going to be landmark study in the food policy design in the UK and encourage similar research elsewhere. The only suggestion I have for the authors to explain the feasibility of observing changes in the health outcomes at 2-3 months interval as proposed. It would useful to explain which of the outcomes can easily change in the short-term while some are slower to respond to diet changes. This would not only increase credibility of the estimated impacts but also highlight how various health outcomes respond to the proposed intervention.
--

VERSION 1 – AUTHOR RESPONSE

Reviewer 1

UPDATE trial: Investigating the effects of Ultra-Processed versus minimally processed Diets following UK dietary guidance on health outcomes: a protocol for an 8-week community-based crossover randomised controlled trial in people with overweight or obesity, followed by a 6-month behavioural intervention- REVIEW

We thank the reviewer for their detailed comments, questions and helpful suggestions. We have provided responses below, and amended the manuscript where necessary.

Comment 1:

Abstract

p. 3 Line 11. 'No studies have assessed whether the health impact of dietary guidelines depends on the extent and purpose of processing, nor provided behavioural support to UK adults with overweight/obesity to reduce their UPF intake.' This sentence seems incomplete and unclear.

Thank you for your comment. We have amended this sentence to page 54-57:

"No study has assessed whether the health impact of adhering to dietary guidelines depends on food processing. In addition, our study is the first to assess the impact of providing a 6-month behavioural support programme aimed at reducing UPF intake in people with a high UPF intake with overweight/obesity."

Comment 2:

p. 3 Line 17. 'Fifty-five adults aged ≥ 18 , living with overweight/obesity (≥ 25 to < 40 kg/m²), with $\geq 50\%$ of habitual energy intake from UPFs will receive an 8-week UPF diet and an 8-week minimally processed food (MPF) diet, both following EWG recommendations, in a random order, with a 4-week washout period.' I suggest mentioning that diets will be delivered to participants' homes.

We thank the reviewer for this suggestion and we have modified the sentence accordingly (line 60-63):

"Fifty-five adults aged ≥ 18 , living with overweight/obesity (≥ 25 to < 40 kg/m²), with $\geq 50\%$ of habitual energy intake from UPFs will receive an 8-week UPF diet and an 8-week minimally processed food (MPF) diet delivered to their home, both following EWG recommendations, in a random order, with a 4-week washout period."

Comment 3:

p. 3 Line 23. 'The primary outcome is the difference in weight change between UPF and MPF diets.' Is the primary outcome the 'difference in weight change between UPF and MPF diets' or 'the weight change between UPF and MPF diets'? Is this at 8 weeks or at both 8 weeks and at 6-months follow-up? If at 6-months follow-up, what (measured when?) would be the initial weight used as a comparator for the 6-months weight? How and when would UPF and MPF diets be characterized?

Thank you for your comment. We apologise that our primary and secondary outcomes were not clear. The primary outcome is the change in weight from baseline to 8-weeks on the MPF diet versus the change in weight from baseline to 8-weeks on the UPF diet, for each participant. This change score will be summed across all participants (within-participant comparison). The brevity is due to the word count in the abstract.

We have now added detail for clarity:

Line 65-66:

"The primary outcome is the difference in weight change between UPF and MPF diets from baseline to week 8."

Weight change at 6 months is a secondary outcome. There are no UPF or MPF diets for the behavioural support programme. Weight change at 6 months is just the difference from baseline to 49 weeks, and is not a comparison of UPF or MPF diets. This is detailed in the manuscript.

Comment 4:

p. 3 Line 24. 'Secondary outcomes include changes in diet, waist circumference, body composition, heart rate, blood pressure, cardiometabolic risk factors, appetite regulation, sleep quality, physical activity levels, physical function and strength, wellbeing and aspects of behaviour change and eating behaviour at 8 weeks between UPF and MPF diets, and at 6-months follow-up.' For 6-months follow-up assessments, what (measured when?) initial outcome assessments would be used as a comparator and how (and when?) would UPF and MPF diets be characterized?

Thank you for your comment. The baseline comparator for the behavioural support programme will be the first baseline measurements. The 6-months follow-up assessments will be compared to the first baseline assessment. This is outlined in the manuscript in paragraph 397-402, but omitted to meet the word count in the abstract. There are no UPF or MPF diets for the behavioural support programme therefore these will not be characterized for the support programme.

Comment 5:

Introduction.

p. 4 Line 37. 'In the only randomised trial to date examining the effect of energy-matched UPF versus MPF diets participants gained nearly 1kg body weight on the 2-week UPF diet, but lost nearly 1kg on the 2-week MPF diet, with over 500kcal/day differences in energy intake²¹. The diets were also matched for presented energy density, carbohydrate, sugar, fat, sodium and fibre content. The MPF diet also led to favourable changes in appetite-regulating gut hormones compared with the UPF diet. This further suggests that the adverse effects of high UPF intakes are independent of individual nutrients such as fat, salt and sugar.' Because it was the offered/presented diets that were matched for nutrients rather than the actual consumed diets, I do not think the matching can disprove that nutrients such as fat, salt and sugar played a role in the weight gain observed during the UPF diet. Did participants consume the same amount of fat, salt and sugar during the UPF and MPF diets? Only if so, might we conclude that nutrient consumption was not a driver of weight gain.

Thank you for this comment – the question is whether to consider the presented nutrients as the driver, or the actual consumption itself. Provision of diets matched for energy intake at the recommended energy requirements for each participant, would be unlikely to lead to differences in weight. We are providing the diets ad libitum as this mimics the real world and allows people to eat to their appetite and reach satiation, to examine the influence of UPF on weight maintenance. Please note that the diets are matched for calories in terms of total amount of food (4000kcal/meal), the total energy intake and nutrient intake will depend on how much they consume of the individual diets, which previous research has shown differs between UPF and MPF, therefore the total amount of the food will impact directly on weight gain.

We have updated the manuscript to reflect your comment, omitting the sentence:

Line 157:

"This further suggests that the adverse effects of high UPF intakes are independent of individual nutrients such as fat, salt and sugar."

Comment 6:

p. 4 Line 56. 'Adopting a healthy diet still presents a significant behaviour change and lifestyle modification which can be challenging for many individuals.' Should 'presents' be replaced by 'requires'?

Thank you for your comment. We have changed the wording, thank you (line 165):

“Adopting a healthy diet requires a significant behaviour change and lifestyle modification, which can be challenging for many individuals.”

Comment 7:

Objectives and hypotheses

p. 5 Line 29 ‘We hypothesise that there will be a difference in the change in weight and other health measures between the two diets; whereby consuming an ad libitum MPF diet complying with the EWG will result in weight loss and favourable changes in cardiometabolic, behavioural, mental and hormonal outcomes, whereas consuming an ad libitum UPF diet complying with the EWG will result in no change in weight or cardiometabolic, behavioural, mental and hormonal outcomes.’ Do you hypothesise that ‘there will be a difference in the change in weight and other health measures between the two diets’ or that ‘there will be a change in weight and other health measures between the two diets’?

Thank you for your comment. As described in lines 194-208, we hypothesise that there will be a difference in the change in weight and other health measures between the two diets, between baseline and 8-weeks follow-up.

Comment 8:

Why do you hypothesise that consuming an ad libitum UPF diet complying with the EWG will result in no change in weight or cardiometabolic, behavioural, mental and hormonal outcomes? Why don't you expect lower weight loss and favourable changes in cardiometabolic, behavioural, mental and hormonal outcomes instead of null effects (if ad libitum UPF diet complying with the EWG has better nutritional qualities than the UPF diet likely less compliant with the EWG before entering the trial)?

Thank you for your comment. Participants eligible for this trial are living with overweight or obesity, and have a relatively high UPF intake. As the study is based in London, an intake above 50% is roughly in line with South England (including London) intakes (<https://www.ncbi.nlm.nih.gov/pmc/articles/PMC7946062/>). Therefore, such participants would not be reducing their UPF intake, and for most individuals, would slightly increase (up to 95% of energy intake).

No weight change was expected on the UPF diet, as a result of following the EWG, which would be expected to be a nutritional improvement on their current diet. But, as this was provided predominantly through UPF and likely an increase on their current intake, would mitigate the expected benefits of following the dietary guidelines.

This approach was used to capture the minimal clinically significant difference between diets. Unlike Hall et al., the UPF diet was adherent to dietary recommendations, rather than just being matched with a control diet, therefore there was less expectation for weight gain with this diet. If of course the diet did result in any weight gain, our power calculation approach ensures sufficient statistical power to observe such differences.

Comment 9:

Methods and analysis

p. 9 Line 11. ‘Participants will complete two non-consecutive 24-hour recalls at screening. If they meet all inclusion criteria and do not meet any exclusion criteria, they will complete a further two nonconsecutive 24-hour recalls at baseline to confirm a habitual dietary intake of $\geq 50\%$ UPF. After which, they will be considered eligible.’ How exactly will you handle the four 24-hour recalls for confirming a habitual dietary intake of $\geq 50\%$ UPF? Will means across measures be calculated?

Thank you for this comment, we will check for a habitual dietary intake of $\geq 50\%$ UPF by taking a mean across all four nonconsecutive 24-hour recalls at screening and baseline.

We have clarified this in the manuscript (line 276-277):

“The mean of all four recalls will be used to determine final habitual UPF intake.”

Comment 10:

p. 10 Line 13. ‘Alcohol will not be provided. Participants will be asked to keep alcohol consumption within government guidelines (14 units per week)³⁴.’ Will participants be instructed to stick their diets to the delivered foods? If so, wouldn’t allowing for alcohol consumption escape from this instruction?

Thank you for your comment. Whether to allow participants to consume alcohol is ultimately a balance between ensuring adequate recruitment, ensuring dietary adherence and reducing participant burden. Previous free-living controlled feeding trials have allowed participants to consume alcohol, and some have even allowed meals off diet to be allowed (<https://www.sciencedirect.com/science/article/abs/pii/S00282239392046Z> and <https://www.sciencedirect.com/science/article/pii/S0002822305006425?via%3Dihub>). We have not allowed participants to have any meals off diet, unlike previous studies. To mitigate against any impact of excessive alcohol consumption, screened participants were excluded if they were high alcohol drinkers consuming above government guidelines and were unable to stick within recommendations, meaning all participants have generally low intakes. For high alcohol drinkers, abstaining from alcohol to participate may present an additional confounding factor at baseline, influencing the results.

As a crossover randomised controlled trial, participants act as their own control. As eligible participants are within government drinking recommendations and have consistent intakes (they are not binge drinkers), intakes will be generally consistent across both diets to allow for estimation of the effects of the diets. We can confirm alcohol intake through questionnaires across the trial, and also in the provided food diaries, where participants will record any alcohol intake. Therefore, we will be able to assess alcohol intake, and adjust for it where necessary.

Comment 11:

p. 10 Line 16. ‘Menu guides will be provided with instructions and pictures to prepare each meal.’ Doesn’t this exclusively apply to MPF diets? If so, shouldn’t this be mentioned?

Thank you for your comment. In planning the menu, we ensured that the practical logistics of both diets were similar, and required minimal effort for participants to make and prepare their diets to maximise adherence and minimise barriers. Our caterer hand-prepared the MPF meals, such that they were ready to eat, or needed a few minutes in the microwave. The MPF diet was not provided as ingredients with a long recipe to follow. Similarly, the UPF diet was constructed from food and drink readily available from leading UK supermarkets, such that they were also ready to eat, or needing a few minutes in the microwave. Therefore, both menus were provided with basic instructions and pictures, and were very similar in presentation and design. These instructions were simple, stating on what days and when to have each meal/snack, and generally just outlining how to heat the meal up, or to just consume as is.

Comment 12:

p. 12 Line 32. ‘Enrolled participants will be block randomised (stratified by night-shift status, sex and ethnicity as potential treatment modifiers) to receive: (1) the MPF diet then UPF diet, or (2) UPF diet then MPF diet.’ Wouldn’t stratification by age also make sense?

Thank you for this comment. We discussed the variables required for stratification through detailed discussion amongst the expert co-authors in obesity and endocrinology for this manuscript, and with the trial statisticians. Night-shift status, sex and ethnicity variables were prioritised as stratification variables. A fourth stratifying variable with age was considered. However, adding a fourth stratification variable would greatly increase the risk of imbalance, creating 16 different combinations of randomization factors for 55 participants. We were advised against doing so. Therefore, we chose to include age as an adjustment covariate in the final statistical models instead.

We have added this detail to the manuscript (lines 694):

“Age will also be included as an adjustment covariate.”

Comment 13:

p. 15 Line 9. ‘Intake24 is a validated, online, self-reported 24-hour recall system, based on a multiple-pass recall suitable for the general population^{73,74} (<https://intake24.co.uk>). The web-based recall method is convenient, efficient and ensures coding consistency. Participants enter all food and drink consumed in the previous 24 hours (from waking up to going to sleep) into Intake24 on two non-consecutive days per visit period. Each recall takes roughly 12 minutes to complete⁷⁵. The first recall at screening will be conducted with the research team to ensure adequate training. Links will then be sent to participants to complete further recalls remotely. Intake24 is connected to the National Diet and Nutrition Survey Nutrient Databank to provide nutrient outputs, and has been coded into NOVA by the research team for calculating UPF/MPF intake.⁷⁶’ Does the Intake24 collect brand name information (important to determine Nova classification) and will brand name information be taken into account when assigning Nova groups?

Thank you for your comment. Yes, the brand names are collected and can be determined from Intake24, and were taken into account when assigning NOVA groups. We have published a preprint on our coding and the process, which has been accepted for publication (<https://www.medrxiv.org/content/10.1101/2023.04.24.23289024v1>)

We have added this detail to the manuscript (line 548-549):

“Intake24 includes details on the brand names of products, facilitating assignment of NOVA groups.”

Comment 14:

p. 15 Line 23. ‘The European Prospective Investigation into Cancer and Nutrition (EPIC)-Norfolk FFQ is a validated, semi-quantitative measure of average dietary intake over the past year^{77–79}. The FFQ contains a 130-item food list followed by detailed questions regarding the food list. For each food item, participants tick the most appropriate frequency of consumption of that item from nine options (from never or less than once per month, to 6+ per day). Participants will complete an FFQ and two non-consecutive day 24-hour dietary recalls at baseline, and 4- and 8-weeks for both diets, and at the 6-month behavioural intervention follow-up (Figure 1)⁸⁰.’ Will the same Nova coding used in previously published EPIC studies be applied to FFQ food items? If FFQ relates to dietary intake over the past year, will it be applied several times to handle potential inconsistencies?

Thank you for your comment. UPF intake will be primarily determined from the multiple, non-consecutive day, 24-hour recalls. We are using the EPIC-Norfolk FFQ as a supporting dietary assessment tool, given the lower detail for coding into NOVA from FFQs, and higher risk of error. The FFQ is assessed once at baseline, and then again at each follow-up visit. The EPIC-Norfolk FFQ has been independently coded into NOVA by the study team who also coded the 24-hour recall database

into NOVA (<https://www.medrxiv.org/content/10.1101/2023.04.24.23289024v1>). This was performed to align coding with the typical foods consumed in the UK as of 2023, to ensure relevance. As a supporting dietary assessment tool, a sensitivity analysis can be conducted with the FFQ, rerunning the estimate of UPF from the FFQ using the EPIC coding. This could be discussed in the future, and would require collaboration with the EPIC nutrition team and published in a subsequent publication. Furthermore, the use of repeated non-consecutive 24-hour recall in addition to FFQ have been suggested to give better dietary recall data than using only recall data.

Comment 15:

p. 15 Line 23. 'IBDA will also be completed on two non-consecutive days during weeks 4 and 8 of the RCT, whereby participants will take photos before and after their ad libitum meals/snacks.' Why isn't IBDA also applied t baseline and at 6-month follow-up?

Thank you for you comment. The IBDA is focussed on assessing intake of the food and drink provided to participants during the RCT, and hence at weeks 4 and 8 only. The IBDA is largely an adherence tool to check that participants are preparing and consuming their provided diets as planned. IBDA will be assessed as a supporting exploratory analysis to see how energy and nutrient intake from the IBDA differs to measures of self-reported energy intake.

Comment 16:

p. 15 Line 23. 'Questionnaires summary. There are twelve questionnaires/surveys: PSQI, IPAQ-SF, EQ-5D-3L, IWQOL-Lite, WEMWBS, PHQ9, GAD7, PoF, CoEQ, COM-B healthy eating and PA questionnaire, Intake24 and EPICNorfolk FFQ. Time points for completion are detailed in Table 3 and in Supplementary Material 4.' Will means across measures be calculated to obtain summary measures?

Thank you for you comment. This is a crossover trial, so we are interested in within-person changes in these questionnaires. Averages and distributions will be used (mean and standard deviation, or median and interquartile range where appropriate) and summed across participants for summary measures at each timepoint. These timepoints will then be compared. For questionnaires with validated cut-offs for severity (e.g. WEMWBS, PHQ9 and GAD7), we will also look at categorical changes in severity across the diets.

Comment 17:

p. 18 Line 35. 'Where possible, participants will be selected to ensure representation from across ethnicities, night-shift patterns genders, treatment allocation arms.' Why not age, also?

Thank you for you comment. As discussed in comment 12, age will be added to the final statistical models to test for an effect of age.

We discussed the variables required for stratification through detailed discussion amongst the expert co-authors in obesity and endocrinology for this manuscript, and with the trial statisticians. Night-shift status, sex and ethnicity variables were prioritised as stratification variables. A fourth stratifying variable with age was considered. However, adding a fourth stratification variable would greatly increase the risk of imbalance, creating 16 different combinations of randomization factors for 55 participants. We were advised against doing so. Therefore, we chose to include age as an adjustment covariate in the final statistical models instead.

Reviewer: 2

Dr. Andaleeb Rahman , Cornell University

Comments to the Author:

Ultra-processed food are increasingly believed to be the leading cause of obesity and associated non-communicable diseases in the world. In UK, Eatwell Guide (EWG) has been prepared by the government to promote a health diet which is low in sugar and contains reduced amount of fat. Yet, the EWG guidelines do not discourage the consumption of UPFs as some of them--with better macronutrients and reduced salt and fat content--are marketed as 'healthy'. This proposed research claims to be the first randomized trial (RCT) study--the gold standard in causal methods--which compares the health effect of processed food. It aims to compare the health outcomes (weight being the primary outcome along with cardiometabolic, behavioral, mental and hormonal difference) of individuals who consumer the similar food in accordance with the EWG recommendations but the diets differ in terms of their processing levels -- Ultra-processed food (UPFs) versus minimally processed food (MPFs).

I feel that the authors have an impeccable understanding of the clinical research methods proposed to be employed here. I am extremely satisfied with the protocol presented here which covers all the bases. The sampling strategy and research design seems solid and I don't think that the small sample size would be stumbling block for statistical inference here. I believe that this is going to be landmark study in the food policy design in the UK and encourage similar research elsewhere.

We thank the reviewer for their supportive comments and suggestions.

Comment 1:

The only suggestion I have for the authors to explain the feasibility of observing changes in the health outcomes at 2-3 months interval as proposed. It would useful to explain which of the outcomes can easily change in the short-term while some are slower to respond to diet changes. This would not only increase credibility of the estimated impacts but also highlight how various health outcomes respond to the proposed intervention.

Thank you for you comment. For the primary outcome, weight can change quickly in response to a dietary intervention, as well as gut-hormone changes, as evidenced by Hall et al, where both weight and gut hormone changes were observed after only 2 weeks. We have also been careful in our choice of secondary outcomes to select those, which are likely to observe changes within the time frame of both diets. Thus, if our hypothesis is supported by the results, changes may be seen across all outcomes. Our choice of secondary measures was chosen based on existing evidence regarding the influence of weight change on other health parameters, as well as research gaps between diet and health. Notable changes can be seen across the range of secondary outcomes reported in this manuscript resulting from diet changes. A number of these scale with the extent of weight loss; weight change directly relates to improvements in cardiometabolic risk factors (e.g., blood pressure, blood glucose, glycated haemoglobin (HbA1c), lipids), physical function and quality of life. Analyses of all secondary measures will be treated as exploratory.

Preliminary evidence from several co-authors of this manuscript and trial (<https://www.bbc.co.uk/programmes/m000wgcd>) showed changes in MRI brain connectivity after only 28 days on a UPF diet.

Evidence regarding the influence of diet on cognitive and behavioural outcomes has been less well studied, and all secondary measures are treated as exploratory. We have maximised the reliability of our results by utilising validated measurement tools throughout all of our secondary outcomes.

We have amended the manuscript to describe the expected changes in outcomes:

Line 366-370:

“This outcome is currently being used clinically in weight management clinics and used for all NHS weight management programmes. Clinically significant weight change can occur from short-term dietary interventions and directly relates to improvements in cardiometabolic risk factors (e.g., blood pressure, blood glucose, glycated haemoglobin (HbA1c), lipids)⁴², physical function and quality of life⁴³.”